# Understanding the structure of cognitive noise

**Jian-Qiao Zhu** [1]*, **Pablo León-Villagrá**[1¤], **Nick Chater**[2], **Adam N. Sanborn**[1]

**1** Department of Psychology, University of Warwick, Coventry, United Kingdom, **2** Warwick Business School, University of Warwick, Coventry, United Kingdom

¤ Current address: Cognitive, Linguistic & Psychological Sciences, Brown University, Providence, Rhode Island, United States of America

* jianqiao.zhu@warwick.ac.uk

**Data Availability Statement:** We have made the experimental data available here: https://osf.io/kcfgp/.

**Funding:** J.Q.Z., P.L-V., and A.N.S. were supported by the European Research Council (ERC; 817492-

## Abstract

Human cognition is fundamentally noisy. While routinely regarded as a nuisance in experimental investigation, the few studies investigating properties of cognitive noise have found surprising structure. A first line of research has shown that inter-response-time distributions are heavy-tailed. That is, response times between subsequent trials usually change only a small amount, but with occasional large changes. A second, separate, line of research has found that participants' estimates and response times both exhibit long-range autocorrelations (i.e., $1/f$ noise). Thus, each judgment and response time not only depends on its immediate predecessor but also on many previous responses. These two lines of research use different tasks and have distinct theoretical explanations: models that account for heavy-tailed response times do not predict $1/f$ autocorrelations and vice versa. Here, we find that $1/f$ noise and heavy-tailed response distributions co-occur in both types of tasks. We also show that a statistical sampling algorithm, developed to deal with patchy environments, generates both heavy-tailed distributions and $1/f$ noise, suggesting that cognitive noise may be a functional adaptation to dealing with a complex world.

## Author summary

Human behavior is inherently noisy, but this noise is surprisingly structured. Moment-by-moment fluctuations in responses are usually small but also occasionally quite large, mirroring a pattern seen in animal foraging. Separate work using different tasks has shown that response fluctuations do not depend just on recent responses, but also on a long history of past responses. In two experiments using very different tasks, we found that these two features co-occur. We show that a particular kind of algorithm, developed in computer science and statistics to approximate answers to difficult probabilistic problems, exhibits both these features as well, suggesting that noise is functionally important.

SAMPLING). The funders had no role in study design, data collection and analysis, decision to publish, or preparation of the manuscript.

**Competing interests:** The authors declare no competing interests.

## Introduction

Human cognition is fundamentally noisy across all kinds of judgments and behaviors [1–3]. In empirical research, noise (often treated as residuals in experimental inquiry) is generally assumed to be a random fluctuation independent of the underlying signal and previous trials, hence a nuisance variable, which is removed by averaging results or counterbalancing experimental designs. Therefore, these noisy residual fluctuations were typically assumed to play no functional role in cognitive tasks, and so in models of cognition they are very often characterized as independent draws from a Gaussian distribution. However, studies investigating the properties of noise in human cognition have instead found interesting structure [3–6].

First, while continuous responses are usually assumed to be normally distributed, it has been found that, in free recall tasks, inter-response intervals (IRIs) follow heavy-tailed distributions. Participants asked to recall animal names (see Fig 1A), mostly produced short intervals between retrievals of animal names, but infrequently their retrieval intervals were much longer. These heavy-tailed distributions of retrieval times, *l*, were well described as power laws: $P(l) \sim l^{-\mu}$ with many participants exhibiting tail exponents of $\mu \approx 2$ [4]. Interestingly, research on animal foraging suggests that the mobility patterns of a wide array of species also exhibit the same exponents, which are mathematically optimal for blind search in environments in which resources are clumped together [7]. Together, these results suggest that human memory retrieval amounts to foraging in a patchy psychological space [6].

Second, when participants make repeated estimates of a temporal duration or a spatial magnitude, their responses are not independent, nor only dependent on the last estimate as would be predicted by a random-walk model. Instead, they also depend on long-ago estimates [3].

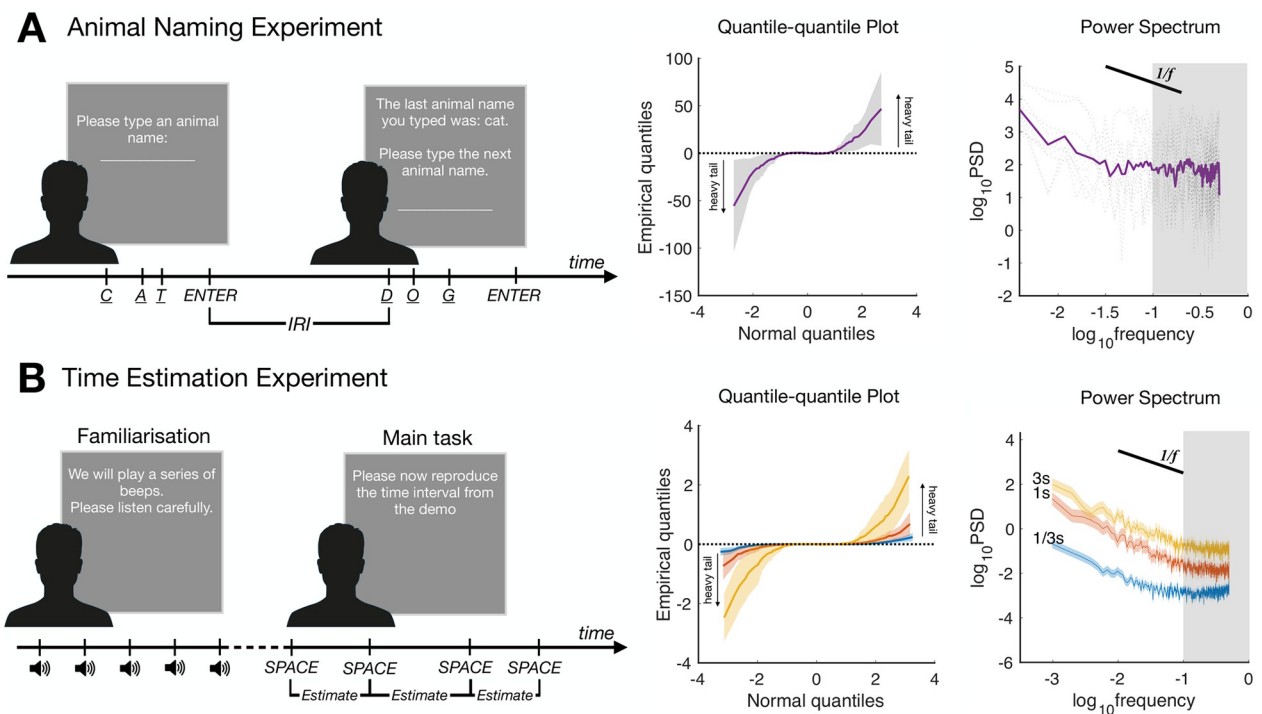

**Fig 1. Experiments and results.** (A) Animal naming experiment. (B) Time estimation experiment. Within each row from left to right, there are illustrations of the experimental procedure, quantile-quantile plots of successive changes in IRI or time estimates to test for heavy tails (95% CI shaded around means and the horizontal dashed lines denoting the Gaussian distribution), and power spectra to test for 1/*f* noise (fitted at frequencies less than 0.1, 95% CI shaded around means).

Similar long-range autocorrelations also occur in the response times of many other cognitive tasks such as mental rotation [5]. These correlations are best described as $1/f$ noise and can be expressed in the frequency domain, $S(f) \sim 1/f^\alpha$, where $f$ is frequency, $S(f)$ is spectral power, and exponents $\alpha \in [0.5, 1.5]$ are considered $1/f$ scaling [3, 5]. $1/f$ noise has been explained as the result of complex organisms displaying a self-organized criticality, meaning the organisms are inherently and continuously transitioning between different stable states and as a consequence exhibit complex behavior [8]. It also appears to be a signature of active cognitive processing, because it disappears if participants are not asked to do anything more than simply press a button in response to a randomly-occurring target [3].

It is very rare to find a cognitive model that predicts either heavy-tailed distributions in trial-by-trial changes or $1/f$ noise. And, because these two effects have been largely studied in isolation, as a consequence, even the small number of models that predict $1/f$ noise do not predict heavy tails and vice-versa. It is not trivial to produce $1/f$ noise and the most common descriptive model, fractional Brownian motion, predicts a Gaussian distribution of successive changes instead of a heavy-tailed distribution [9]. Conversely, the most common model of heavy tails in successive changes, the Lévy flight, is a random-walk model that does not produce long-range autocorrelations [7]. In sum, standard models suggest that heavy-tailed distribution by themselves do not simply imply $1/f$ noise and vice-versa.

Here, we investigate whether $1/f$ noise and heavy-tailed distributions of trial-by-trial changes co-occur in the same experimental task. We ran two experiments with very different tasks: an animal naming task, previously used to show heavy tails [4], and a time estimation task, previously used to show $1/f$ noise [3]. In the animal naming experiment, participants were instructed to type animal names as they came to mind, with the only constraint being that successive names needed to be different. In the time estimation experiment, participants were first presented a demonstration of a target time interval (1/3, 1, or 3 seconds) and then were asked to repeatedly reproduce the interval, as if they were drumming (see Fig 1 and Materials and Methods for details).

To determine whether trial-by-trial changes followed a heavy-tailed distribution, we fit $\mu$ following a similar procedure to [10], and also used stricter tests based on directly comparing heavy and exponential tails [11] that show qualitatively very similar results (see Text A in S1 Supporting information). To measure the autocorrelation exponent $\alpha$, we fit a line to the windowed, log-binned power spectrum for low frequencies (i.e., less than 0.1) following [3].

## Results

In the animal naming experiment, both the pooled data (see Fig 1A) and all individual exponents indicate heavy-tailed distributions ($\hat{\mu}$ in the range of [1.29, 2.61]), which lies within the (1, 3] range indicating heavy tails, replicating [4]. Congruent with the idea in memory foraging, participants were more likely to report successive animal names that belonged to the same category (e.g., patch) than other categories. In addition, IRIs were longer when transitioning between categories, and IRIs were correlated with distance in a semantic embedding (see Text A in S1 Supporting information). Finally and critically, these data also show $1/f$ noise, both for the pooled results ($\hat{\alpha} = 1.08$) and for 9 of 10 individual participants ($\hat{\alpha}$ in the range of [0.38, 0.90]).

Our time estimation experiment also found $1/f$ noise, replicating [3]. For the pooled data, $\hat{\alpha}$ was 1.10, 1.45, 1.27 for the 1/3s, 1s, and 3s conditions, respectively. 21 of 30 participants had exponents in the $1/f$ range ($\hat{\alpha}$ in the range of [0.53, 2.06]). Importantly, the time estimation experiment also showed heavy-tailed trial-by-trial changes in the pooled data (see Fig 1B), and

in 23 of 30 participants ($\hat{\mu}$ in the range of [0.87, 2.67]). We present evidence that this finding was not due to lapses of attention in Text A in S1 Supporting information.

Overall, in both experiments, heavy-tailed trial-by-trial changes and $1/f$ noise co-occurred, and many individuals exhibited convincing evidence of both effects: 9 of 10 participants in the animal naming experiment and 17 of 30 participants in the time estimation experiment.

The co-occurrence of heavy tails and $1/f$ noise invalidates the most common accounts for each and calls for another explanation. One possible direction is to describe noise using a more complex statistical process, such as Brownian motion in multifractal time, that can, for some parameter settings, produce both features of cognitive noise [12]. However, this account is incomplete as it does not explain how people are able to perform the time estimation task—it is silent on why participants' average estimates tracked the target times ($M = 0.36$s for 1/3s, $M = 1.19$s for 1s, and $M = 3.50$s for 3s). That is, descriptive models of noise offer a very incomplete account of human behavior.

An interesting alternative that does explain performance casts the mind as an intuitive statistician: the brain creates probabilistic models of an uncertain world and acts according to the prescriptions of these models when taking action [13, 14]. These models have successfully explained many aspects of effortful cognitive tasks, which, like our animal naming task, require memory retrieval [15], as well as automatic-seeming perceptual tasks such as time estimation [16, 17]. However, they have been criticized on the grounds that they are far too complex to be psychologically plausible, and on the grounds that people show systematic deviations from Bayesian statistical models.

Statisticians address the intractability of probabilistic models by using approximations such as sampling algorithms, and it could be that the brain does the same: it utilizes sampling algorithms similar to those used in statistics to approximate otherwise intractable solutions. Sampling approximations are appealing as they show many of the same deviations from exact probabilistic inference that people do [18], and provide an explanation for behavioral and neural variability [19, 20].

There are many types of sampling algorithms and the simplest, drawing independent samples, requires knowing the probability of every hypothesis. This itself is often computationally intractable, leading to the development of a sophisticated family of sampling algorithms called Markov Chain Monte Carlo (MCMC) [21]. MCMC algorithms traverse the space of hypotheses using only local information about the probability distribution. These local transitions mean that successive samples are unavoidably autocorrelated, a necessary evil as these samples very often convey less information than the same number of independent samples do.

The ability to operate with only local information about a probability distribution is the strength of MCMC algorithms, but it introduces weaknesses. Multi-modal probability distributions, those in which there are multiple clusters of high-probability hypotheses which are separated by regions of low-probability hypotheses, are a challenge to these algorithms. Multi-modal probability distributions formalize the idea of patchy mental representations, such as those that researchers assume participants use in animal naming experiments [4].

Having only local knowledge, MCMC algorithms are often stuck in one patch and require a large number of iterations before they can visit other isolated modes. As illustrated in Fig 2, this is a problem both for basic MCMC algorithms such as Random Walk Metropolis (RWM), and more advanced versions such as Hamiltonian Monte Carlo (HMC). To address these weaknesses, statisticians have developed MCMC algorithms that deal better with multimodal representations. One of the first, Metropolis-coupled MCMC (MC$^3$), runs multiple MCMC chains in parallel: while one chain produces the samples, the remaining chains explore the space for isolated modes [22]. When an exploring chain finds an isolated mode, the sampling chain is likely to swap positions with the exploring chain, and then start sampling from that

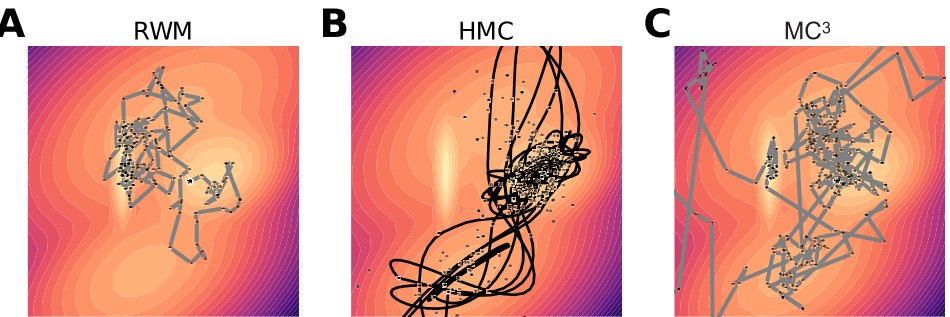

**Fig 2. Example trajectories for the three samplers for a mixture of multivariate Gaussians.** (A) Random Walk Metropolis (RWM), (B) Hamiltonian Monte Carlo (HMC), and (C) Metropolis-coupled MCMC (MC³).

isolated mode (see Fig 2 and descriptions in Text B in S1 Supporting information). The interaction between chains is crucial for MC³ to switch locations between modes (assuming an exploratory chain has reached another mode). However, there are foraging rules that explore patchy environments with only knowledge about the mode that the agent is situated in (see [23, 24] for detail). For these foraging rules, switching to a new mode often requires random exploration, as is also the case for both RWM and HMC. Nonetheless, there are potential overlaps between foraging rules and sampling algorithms as any foraging rule that visits locations according to their probabilities is a sampling algorithm. Therefore, foraging rules with memory that can produce autocorrelations are interesting candidates for future investigation.

Past work has not compared how well these different MCMC algorithms match human behavior, most likely because they are difficult to distinguish in most experimental data, and time-series data is needed to do so. Neither $1/f$ noise nor heavy-tailed changes naturally arise from any of these algorithms, and in particular, $1/f$ noise will be the result of a mixture of processes with specific scales, and will show a power-law relationship only within a range. There is no reason to expect a priori that the samplers will emit both heavy-tailed changes and $1/f$ noise.

We quantitatively investigated RWM, HMC, and MC³ to see which, if any, would produce heavy-tailed distributions and $1/f$ noise. The sampling algorithms were assumed to operate in a hypothesis space, which in the time estimation task was the space of possible time estimates and in the animal naming task was a semantic space of possible animal names. From inspection, the distributions of responses in the time estimation task were unimodal, and so we used a Gaussian distribution as the target distribution, $P(H)$, for all three sampling algorithms in this task, and time estimates were assumed to be direct readouts from the samples of hypotheses, $h_i \sim P(H)$. Animal name IRIs, however, are more complex to model as they require more assumptions about both the target distribution and about how the samples relate to IRIs. The distribution we used (shown in Fig 2) was obtained by fitting a Gaussian mixture model to the animal names arranged in a two-dimensional abstract semantic space in which the sampled points could lie between animal names (see Text C in S1 Supporting information for details). To relate samples to IRIs, we assumed that samples were generated at a constant rate but at random times (i.e., following a Poisson process), and that the animal name "in mind" was the nearest animal name to the current position of the sampler in the semantic space. In line with our experimental instructions that asked participants to report an animal name as soon as the animal name they had in mind changed, IRIs were on average proportional to the number of samples that were generated before the nearest animal name to the sampler's position changed. Further details about these assumptions and an exploration of an alternative in which IRIs are

**Table 1. Model scores for the three sampling algorithms: Random Walk Metropolis (RWM), Hamiltonian Monte Carlo (HMC), and Metropolis-coupled Markov Chain Monte Carlo ($MC^3$).**

| Experiment | Model Scores | RWM | HMC | $MC^3$ |
|---|---|---|---|---|
| Animal Naming | log marginal likelihood | -16.0844 | -29.5261 | -16.8500 |
| | number of best-fitting participants | 7 | 0 | 3 |
| | protected exceedance probability | 0.59 | 0.16 | 0.25 |
| Time Estimation | log marginal likelihood | -303.7798 | -301.9607 | -193.0191 |
| | number of best-fitting participants | 0 | 2 | 28 |
| | protected exceedance probability | 0.00 | 0.00 | 1.00 |

related to the distance between samples are given in Text C and Table A in S1 Supporting information.

To assess which sampling algorithm best describes the co-occurrence of heavy tails and autocorrelation observed in human data, we performed a likelihood-free model comparison known as Approximate Bayesian Computation (ABC) [25]. ABC is an approximation to the gold standard measure of marginal likelihood that is particularly suitable for our data because (i) the presence of autocorrelations in time series is notoriously difficult to be properly controlled for with traditional methods, (ii) simulating a sampler's trace is relatively easy, and (iii) the simulated and observed $\hat{\alpha}$ and $\hat{\mu}$ are compared directly. The detailed ABC procedure is given in Text C in S1 Supporting information.

Table 1 summarises the model comparison results for the three sampling algorithms. We find that RWM and $MC^3$ perform similarly overall on the animal naming task with a small advantage for RWM: from the marginal likelihoods RWM was 2.2 times more likely than $MC^3$. Both RWM and $MC^3$ decisively outperformed HMC: both were more than 300,000 times more likely than HMC. As discussed in Text C in S1 Supporting information, HMC performed better when assuming that IRIs were related to the distance the sampler travelled in the semantic space, but the versions of RWM and $MC^3$ presented here still were more than 25 times more likely than the best-performing version of HMC. This, of course, means that conclusions about the sampling algorithms cannot be drawn without also specifying how the samples relate to behavior. RWM is the best fitting model for the largest proportion of participants, but the protected exceedance probability (i.e., the probability that a model describes the greatest number of participants [26]) was unconvincing. In the time estimation experiment, $MC^3$ decisively provides a better account of participant data than either HMC or RWM: it was more than $2 \times 10^{47}$ more likely than either alternative algorithm. $MC^3$ also convincingly fit the largest number of participants, as measured both by raw counts and by protected exceedance probabilities.

## Materials and methods

### Ethic statement

Ethical approval for the experiments was given by the Department of Psychology Research Ethics Committee at the University of Warwick. Written informed consent has been obtained from the participants.

### Animal naming experiment

**Participants.** Ten native English speakers (6 female and 4 male, aged between 19–25 years) were recruited from the SONA subject pool of the University of Warwick.

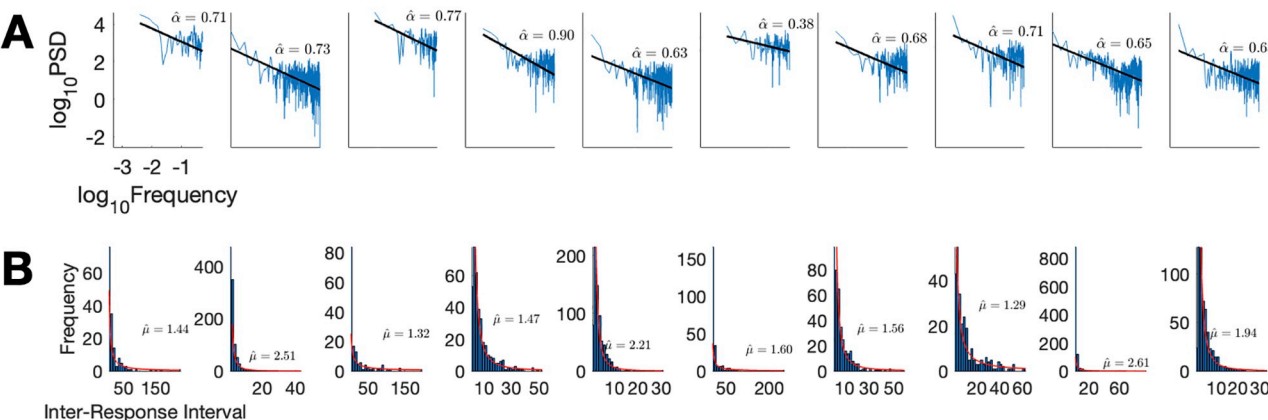

**Fig 3. Individual data for the animal naming experiment.** (A) Power spectral density. (B) Histograms of inter-response intervals. Fitted lines and curves are overlaid with exponents.

**Procedure.** Participants were asked to type animal names as they came to mind and were explicitly instructed that they could resubmit previous animals, though not consecutively. Once participants had typed an animal they submitted their choice by pressing the enter key. The inter-response interval (IRI) between two consecutive animal names was the duration between last enter pressed and the very next key response. The experiment lasted about 60 minutes or until the participants submitted 1024 animals. Participants received £6 for participating.

**Analyses.** IRIs were calculated per-participant for successive submissions. On average it took participants 5.30 seconds ($SD = 11.03$) to submit an animal name. We fitted autocorrelation exponents to the IRI sequences using windowed, log-binning fits [3]. Per-participant correlation exponents ($\alpha$) fits ranged from 0.38 to 0.90, ($M = 0.68$, $SD = 0.12$), and 9 in 10 exponents were in the $1/f$ range. We also obtained per-participant tail exponents ($\mu$) ranging from 1.29 to 2.61 ($M = 1.80$, $SD = 0.49$) (see Fig 3 for details).

## Time estimation experiment

**Participants.** Another 37 participants (13 male, 23 female, 1 undisclosed gender, aged between 18 and 41) were recruited through the SONA subject pool of University of Warwick.

**Procedure.** Participants first listened to a sample of the target temporal interval for 60 seconds, presented as computer-generated beeps. Following this familiarization period, participants were instructed to reproduce the beeps (effectively to estimate the target time interval) by pressing the spacebar when they believed the target interval had elapsed. The next trial began as soon as the last one ended, making the task similar to 'drumming' at the rate of the target interval. The experiment was terminated when participants produced 1030 keystrokes or the maximum experimental duration was reached. Participants were paid relative to the maximum duration of experiment, which varied across the three conditions (6, 20, and 60 mins), receiving £2, £4, and £6 respectively.

**Analyses.** Of the initial sample, 30 participants completed over 512 time estimates, and their data was analysed in the main text. There were 10 participants for each of three target time interval conditions (1/3s, 1s, and 3s). To exclude possible resting periods, we only analysed the time estimates that were less than 3 times the target time interval. Further analyses to exclude possible resting periods are presented in the Text A in S1 Supporting information.

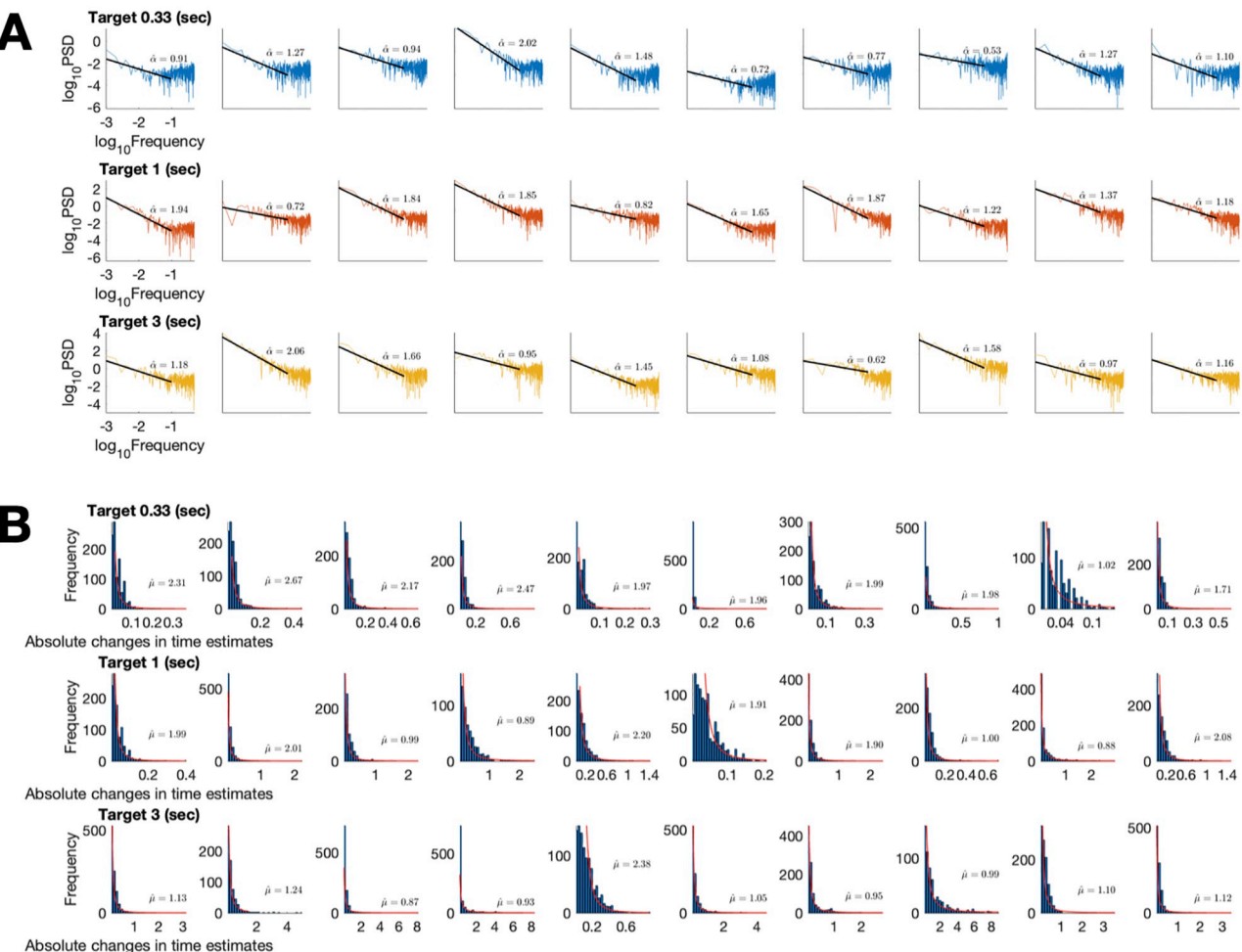

**Fig 4. Individual data for the time estimation experiment.** (A) Power spectral density. (B) Histograms of absolute changes in time estimates. Fitted lines and curves are overlaid with exponents.

Per-participant correlation exponents ($\alpha$) ranged from 0.53 to 2.06 ($M = 1.27$, $SD = 0.44$) and tail exponents ($\mu$) ranged from 0.87 to 2.67 ($M = 1.60$, $SD = 0.59$) (see Fig 4 for details).

## Discussion and conclusions

Very few cognitive models predict either heavy-tailed distributions of trial-by-trial changes or $1/f$ noise, and none of which we are aware predict them both. However, in replicating two standard cognitive tasks, each of which has been independently used to identify either heavy tails or $1/f$ noise, we find a strong evidence for the co-occurrence of heavy tails and $1/f$ noise.

We explored the functional role of heavy tails and $1/f$ noise through the lens of approximations to Bayesian models of cognition. Three sample-based approximations were studied: RWM, HMC, and MC³. In the time estimation task, MC³ better described the human data compared to RWM and HMC in terms of marginal likelihood and number of best-fitting participants. In the animal naming task, MC³ and RWM were comparable on marginal likelihood with the number of best-fitting participants favouring the latter. However, this measure of relative performance does not answer the question of absolute performance: whether MC³ produces the key outcomes of heavy tails and $1/f$ noise. To answer this critical question, we

**Table 2. Correspondence of $1/f$ noise and heavy tails between human data and the MC$^3$ posterior predictive distribution.**

| | | Animal Naming Experiment | | | | Time Estimation Experiment | | | |
|---|---|---|---|---|---|---|---|---|---|
| | | $1/f$ noise | | heavy tails | | $1/f$ noise | | heavy tails | |
| | | yes | no | yes | no | yes | no | yes | no |
| MC$^3$ posterior predictives | yes | 9 | 0 | 10 | 0 | 18 | 0 | 23 | 7 |
| | no | 0 | 1 | 0 | 0 | 3 | 9 | 0 | 0 |

calculated the modal $\alpha$ and modal $\mu$ from the posterior distribution for each participant, and classified each as indicating heavy tails or $1/f$ noise in the same way we did with the experimental participants. We found a perfect correspondence in animal naming and a strong correspondence in time estimation, with MC$^3$ showing slightly less prevalence of $1/f$ noise but a greater prevalence of heavy tails than the human participants (see Table 2).

Overall, we find that MC$^3$ provides a good account of the human data in two very different tasks. This is perhaps less surprising in the animal naming task, as the patchy representation is one for which the algorithm was designed, and complex environments can produce complex behavior from even simple algorithms. It is more surprising in the time estimation task. We speculate that people behave like MC$^3$ in this task because it is a generally useful algorithm, and while it is unnecessary to run multiple Markov chains to sample from a unimodal distribution, it can be difficult to identify when a distribution truly is unimodal: there might always be a distant mode that the sampling algorithm has yet to encounter.

Our finding that $1/f$ noise and heavy-tailed trial-by-trial changes co-occur provides more information about the structure of the variability in human cognition, and is useful in distinguishing between different accounts of noise. While other accounts can describe the co-occurrence, the success of a sampling algorithm in doing so while accomplishing task goals raises the possibility that noisy responses are the signature of a rational approximation in action, rather than a systematic problem with the brain's hardware.

## Supporting information

**S1 Supporting information.** Text A: Data Analysis Methods. Text B: Sampling Algorithms. Text C: Model Comparison Methods. Table A: Model scores for the three sampling algorithms in the animal naming experiment using the alternative method for generating IRIs.
(PDF)

## Acknowledgments

We thank Victoria Eshelby for assisting the research.

## Author Contributions

**Conceptualization:** Jian-Qiao Zhu, Nick Chater, Adam N. Sanborn.

**Data curation:** Jian-Qiao Zhu.

**Formal analysis:** Jian-Qiao Zhu, Pablo León-Villagrá.

**Funding acquisition:** Adam N. Sanborn.

**Investigation:** Jian-Qiao Zhu, Pablo León-Villagrá, Nick Chater.

**Methodology:** Jian-Qiao Zhu, Pablo León-Villagrá, Nick Chater, Adam N. Sanborn.

**Project administration:** Nick Chater, Adam N. Sanborn.

**Software:** Jian-Qiao Zhu, Pablo León-Villagrá.

**Supervision:** Nick Chater, Adam N. Sanborn.

**Validation:** Jian-Qiao Zhu, Pablo León-Villagrá.

**Visualization:** Jian-Qiao Zhu, Pablo León-Villagrá.

**Writing – original draft:** Jian-Qiao Zhu, Adam N. Sanborn.

**Writing – review & editing:** Jian-Qiao Zhu, Pablo León-Villagrá, Nick Chater, Adam N. Sanborn.

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
