## [Decision Letter · Decision Letter 0]

6 Dec 2021

Dear Dr. Zhu,

Thank you very much for submitting your manuscript "Understanding the Structure of Cognitive Noise" for consideration at PLOS Computational Biology.

As with all papers reviewed by the journal, your manuscript was reviewed by members of the editorial board and by several independent reviewers. In light of the reviews (below this email), we would like to invite the resubmission of a significantly-revised version that takes into account the reviewers' comments. Reviewers 1 and 3 are relatively positive, but Reviewer 2 recommends rejection. I find Reviewer 2's arguments fairly compelling, but I also think that you might be able to address them. In addition, I agree with Reviewer 1 that 27%  (while greater than 4% and 2%) is not particularly impressive in absolute terms. I'm not sure that this result provides such compelling evidence for MC3.

We cannot make any decision about publication until we have seen the revised manuscript and your response to the reviewers' comments. Your revised manuscript is also likely to be sent to reviewers for further evaluation.

Sincerely,

Samuel J. Gershman

Deputy Editor

PLOS Computational Biology

Reviewer's Responses to Questions

**Comments to the Authors:**

Reviewer #1: This paper explores two interesting features of human response-time distributions: (a) their heavy-tailed shape and (b) their autocorrelation over long time periods (1/f noise). Although there have been theories that explain each of these features of RTs independently, the authors show that they can co-occur in two disparate cognitive tasks (animal naming and rhythmic time estimation), which suggests a potential common explanation for (a) and (b). In simulations, the authors show that both of these features can arise from a specific type of MCMC algorithm (MC3), which efficiently transitions between different chains when one chain seems to find a novel mode. Thus, if the human brain is implementing a similar algorithm when performing tasks like animal naming, as some cognitive theories suggest, then this provides a parsimonious explanation for both the heavy-tailed RT distribution and 1/f noise.

I enjoyed reading this paper and found the ideas intriguing. Although I believe the work is well-worth being out in the literature, it would be helpful for the authors to fill in some key details in a revision.

Perhaps this is partly explained by my relative unfamiliarity with the subject matter, but I suspect many people would benefit if the authors put in more effort to connect the empirical section of the paper with the simulation work. For example, as of now, I don’t even have a sense of exactly how the authors envision that the time estimation task might use an MCMC algorithm. The task involves rhythmically tapping the space bar at the same tempo for a long period of time. This doesn’t strike me as the kind of cognitively taxing prediction/estimation task that MCMC is typically used for. Moreover, although I have some sense of how MCMC could be relevant for the animal naming task, I’m still not quite sure how to map the MCMC algorithm onto the behavior. For example, when an animal pops into a person’s mind, what is the corresponding ‘event’ in MCMC? Presumably subjects would not be reporting literally every sample from that algorithm, so what constitutes a sample that reaches conscious awareness?

I also worry a bit about the generalizability of the simulation results. Even for the MC3 algorithm, only 27% of parameterizations yielded the combined distributional features that are characteristic of human responses. Is there good reason to believe that these parameterizations — or the MC3 algorithm itself, for that matter — are psychologically plausible? This may be a difficult question to answer given the lack of research on this topic, but more context for how these parameters relate to psychology would be useful for assessing the plausibility of the theory. Furthermore, why was the MCMC algorithm only run on relatively simple probability distributions? Why not model an environment that resembles the more patchy semantic network for animals or other task-relevant domains?

In sum, the theoretical ideas explored in this paper seem quite interesting, but the authors could expand upon the ways in which their simulation models connect to the psychological phenomena of interest and provide more detail about how well the models really capture human behavior.

Minor:

- I’m not sure what this means: “1/f noise has been explained as the result of complex organisms displaying a self-organized criticality” (p. 4).

- In Figure 1 (or another figure), it would be helpful to see some simple histograms of participants’ actual RTs. I don’t have a great sense of how well these distributions are fitted by a power law or exhibit 1/f autocorrelation. Is there any way to directly overlay model predictions on data?

Reviewer #2: The authors show that two human tasks exhibit both heavy tailed distributions of trial-to-trial differences as well as 1/f autocorrelations. They posit that the co-occurrence of these two statistical properties is difficult to explain under standard accounts, and thus might reveal something about the cognitive mechanisms underlying the noise. Finally, they argue that Metropolis-coupled MCMC exhibits the same combination of heavy-tailed, 1/f autocorrelated behavior, suggesting that this is what people are doing. I have two major reservations about this paper: (1) I am not convinced that the combination of heavy-tailed distributions and 1/f autocorrelations is particularly exotic, and (2) i am not convinced that the characterization of MC^3 dynamics is sufficiently diagnostic. Together, these two reservations undermine the major contribution of this paper: the suggestion that the correspondence of these dynamics in human behavior and inference algorithms reveals something about human cognitive mechanisms.

How rare is the co-occurrence of heavy-tailed difference distributions and 1/f autocorrelations? This is the first time I have heard the claim that the co-occurrence is rare, so I went to the obvious place to check: natural images. I downloaded a random grayscale natural image from google image search, pulled out one row of pixels, and evaluated the heavy-tail-ness of the pairwise pixel luminance difference, as well as the log-log slope of the power spectrum. I got both -- the log-log slope of the power spectrum was -0.6, and the log-log slope of the mass in the tails of the distribution was -1.3. So, the first place I looked that has natural 1/f noise also has heavy tails for the adjacent pairwise distances (according to the authors' criteria). If luminance in natural images has this co-occurrence as well, I am not sure what we have to learn about cognitive mechanisms from this co-occurrence. If the authors want to make any kind of argument about cognitive mechanisms from these characteristics of response time distributions, it seems important to show that heavy tailed differences are rare among the many natural phenomena that exhibit 1/f noise (https://en.wikipedia.org/wiki/Pink_noise#Occurrence), because if DNA sequences, quasar light emissions, meteorological data series, are like luminance in natural images and response times, and also exhibit the conjunction of 1/f noise and heavy tailed distributions, then appealing to a particular cognitive mechanism to explain the co-occurrence sounds a bit far-fetched.

My next concern pertains to evaluating whether MC^3 dynamics exhibit these properties. As far as I can tell, the authors fit lines in log-log space to either the power spectrum, or to the tail probabilities of difference distributions, and then adopted certain cutoffs for acceptable slopes. I find the logic of this approach a bit confusing, because one can always fit a line in log-log space. Spectra and tail probabilities will (nearly always) be diminishing, so the slope will be negative, and thus some exponent fitting the tail probabilities or power spectrum will be identified. So even fitting the tail of a normal distribution in this manner will yield a value of mu. The value of mu that will be obtained for non-power-law relationships will depend greatly on the range of values being considered. For instance, in fitting a power law to the tail of a standard normal distribution, I can get an exponent of -2 when considering z-scores of 1 to 2 (when considering z scores from 1 to 5, I get an exponent of -7.5). So a lot of the action in calling these distributions "power law" amounts to details of how the range of the tail is chosen, and what exponent cutoffs are used. Given the somewhat vague criteria for calling some difference distribution heavy-tailed, or some power spectrum 1/f, the pithy summary in Table 1 leaves much to be desired. For instance:

- if one tried really hard in specifying a particular posterior distribution and proposal distributions, is it really the case that random walk metropolis and hamiltonian monte carlo would not be able to meet the criteria for alpha and mu? If some specifications regularly work, but some don't, what does that mean?

- on the flip side, is meeting the alpha and mu criteria roughly half the time really that compelling of evidence that MC^3 exhibits 1/f dynamics and heavy tailed change distributions?

Basically, both the negative, and positive conclusions drawn from Table 1 seem questionable to me, so I'm not sure how much we can conclude based on the differences among algorithms.

Altogether, I am left with the impression that (a) the co-occurrence of heavy-tailed distributions and 1/f dynamics is not unique to cognition, so any cognition-specific explanation of the co-occurrence seems overly narrow, and (b) the tests that aimed to evaluate whether particular inference algorithms also exhibited these characteristics seemed not very diagnostic. The net result leaves much to be desired. Moving forward, I would like to see some stronger support for the claim that the conjunction of heavy tails and 1/f dynamics is somehow unique to cognition, and more rigorous evaluations of these properties in candidate algorithms.

Reviewer #3: Review of “Understanding the Structure of Cognitive Noise”, submitted to PLOS Computational Biology (PCOMPBIOL-D-21-01984)

Reviewed by Peter M. Todd, Indiana University Cognitive Science Program

This is a compelling paper that convincingly shows that two types of noise seen in human behavior can systematically co-occur, and demonstrates a cognitive sampling mechanism that can produce that systematicity. The two empirical tasks are appropriate for making these points, along with the comparison between three sampling algorithms to fit the human data. The authors give support for the useful perspective that noise can be “the signature of a rational approximation in action” rather than an indication of cognitive or neural error as others have been arguing. Overall, the paper makes an important contribution on an interesting question with theoretical impact, is well written, and contains sufficient detail to afford replication and further research.

It is good to see two quite different tasks—naming animals and estimating durations—being used to show instances of co-occurrence of the two noise patterns under consideration. Then for the possible underlying mechanism as a sampler over a particular task-related distribution, the applicability of a multimodal distribution to the animal naming task is clear, as people produce multiple clusters of related animal names, but how does it apply to the time estimation task? Is there the expectation in the time literature that an individual would have multiple likely time estimate values (multiple modes in the distribution) that they would be sampling from (rather than one mode)? Or is the point to show that in a task that probably has a unimodal outcome distribution, the Metropolis-coupled MCMC algorithm can still (at least sometimes) show both heavy-tailed and 1/f noise patterns? This should be clarified.

The Metropolis-coupled MCMC algorithm used here switches between patches based on comparing the current patch with another one that has already been discovered, which could possibly be what is happening via parallel search in someone’s mind, but seems unlikely for individual animals foraging among patches in space. Humans also appear to use some patch-switching rules analogous to those used by foraging animals, in some tasks involving mental search (e.g., a giving-up-time rule—see Wilke et al., 2009). It could be interesting to see how this algorithm could work with such patch-switching rules that only use “local” information about the current patch, not about other available patches.

Wilke, A., Todd, P.M., and Hutchinson, J.M.C. (2009). Fishing for the right words: Decision rules for human foraging behavior in external and internal search tasks. Cognitive Science, 33, 497-529.

This question also points to a difference between the two types of tasks used here—in the standard animal naming task, there is the aspect of “using up” the resources in each visited mode, since animal names are supposed to not be repeated, and hence there is in some sense an increasing impetus driving the person to switch from the current mode/patch to a new one (or at least, there’s a benefit to switching at some point—though perhaps not in this paper’s particular version of the task, where repetitions *are* allowed), but in the time estimation task, that impetus does not seem to be present—why should a person (or the sampler) move away from a given mode in this case? (And again, as mentioned above, why expect that there *is* more than one mode in the time estimate distribution?)

Finally, as another way to think about the likelihood of the co-occurrence of heavy tails and 1/f noise, given a particular set of values that are drawn from a heavy-tailed distribution, how likely is any particular sequential ordering of those values to show long-range autocorrelations like 1/f noise? (For example, what is the proportion of scrambled re-orderings of the data from these two tasks that still show 1/f autocorrelation?) Does this differ if the values are produced by a Lévy distribution versus by a non-blind patchy foraging process like area-restricted search operating on particular kinds of patch-size distributions (e.g., the latter could produce long IRIs/durations that are more regularly spaced across a sequence than the former)? If so, could this distinguish the processes going on in the two paradigms here (memory search and time estimation), possibly indicating why all participants showed both patterns in memory search but about half of participants showed both patterns in time estimation?

Detailed comments by page (using manuscript-page numbers):

MS Page 5 top: “Conversely, the most common model of heavy tails in successive changes, the Lévy flight, is a random-walk model that does not produce long-range autocorrelations [7].” –this is presumably because the Lévy flights are assumed to apply to blind foraging over patchy resources with no memory, in contrast to commonly-seen area-restricted search strategies that do involve memory (e.g. making small movements to stay foraging locally as long as resources are plentiful) and hence can produce autocorrelation.

P5 bottom: “all individual exponents indicate heavy-tailed distributions”—indicate what criterion is used to make this assessment (e.g. some particular range of tail exponent mu?). (I assume the criterion for 1/f noise given at the bottom of p. 3 is what is being used for that classification in the paper.)

P5 bottom: “participants were more likely to report sequential animal names that belonged to the same category (e.g., patch) than other categories”—replace “sequential” with “successive” to emphasize that this is about the movement from one animal name to the very next one (not necessarily a longer sequence)?

Pp5-6: the heavy-tail exponent estimates (mu) should be given for both tasks here (also so they can be compared with the mu estimates provided in the Supplementary Text on pp. 16-17).

P13 top: It is surprising to see that participants could resubmit the same animal name repeatedly, which is not typically allowed in this verbal fluency task (e.g., not in Rhodes & Turvey, who specified “without repetition” but still got around 4 repetitions over 20 minutes). Why was that done here? Perhaps to get around the potential problem of “using up” resources in this task, which differs from the time estimation task? (see comment above) How often *did* participants repeat animal names, and did they do so while still in a particular patch, or later after returning to a specific patch?

P18 middle, Algorithm 1: the call to MCMC Step is missing the second argument (of three).

P19 middle, Algorithm 2: in the line “for s = 0 : M//2 do” M is undefined.

P20 middle, Algorithm 3: Why are there even line numbers? In line (4), I is undefined (should it be 1?). And in line (5), “for l:L do”, shouldn’t this be “for j=1:L do” (l is undefined)?

P21 top: “We evaluated power-law exponents and sample autocorrelations for MCMC, MC3, and HMC” – should MCMC be RWM (as on the previous line)?

P22 middle: should “MCMC” everywhere it appears in the last paragraph be “RWM”? And why is MMMD used later in the paragraph when just M is used at the beginning—do these refer to two different values?

**Have the authors made all data and (if applicable) computational code underlying the findings in their manuscript fully available?**

Reviewer #1: Yes

Reviewer #2: Yes

Reviewer #3: **No: **The data is available online, but I did not see the code there (though pseudocode is given in the supplementary text)

PLOS authors have the option to publish the peer review history of their article (what does this mean?). If published, this will include your full peer review and any attached files.

Reviewer #1: No

Reviewer #2: No

Reviewer #3: No
---

## [Decision Letter · Decision Letter 1]

10 May 2022

Dear Dr. Zhu,

Thank you very much for submitting your manuscript "Understanding the Structure of Cognitive Noise" for consideration at PLOS Computational Biology. As with all papers reviewed by the journal, your manuscript was reviewed by members of the editorial board and by several independent reviewers. The reviewers appreciated the attention to an important topic. Based on the reviews, we are likely to accept this manuscript for publication, providing that you modify the manuscript according to the review recommendations.

Sincerely,

Samuel J. Gershman

Deputy Editor

PLOS Computational Biology

[LINK]

Reviewer's Responses to Questions

**Comments to the Authors:**

Reviewer #1: I appreciate the authors' thoughtful responses to my comments and the helpful changes they made to the paper. I have no further concerns and think the paper is ready for publication.

Reviewer #3: Review of revision of “Understanding the Structure of Cognitive Noise”, submitted to PLOS Computational Biology (PCOMPBIOL-D-21-01984-R1)

Reviewed by Peter M. Todd, Indiana University Cognitive Science Program

Overall, the authors have done a thorough and (to me) mostly convincing job of responding to the reviewers’ comments. It is good to see their clarification of their aim to show both that heavy tails and 1/f noise co-occur in some cognitive activities, and that standard cognitive models do not produce this co-occurrence, but MC3 does; a brief summary of this aim would be good to include in the text of the paper as well. It was also very helpful to see that long-tailed distributions by themselves do not imply 1/f patterns—that might also be helpful for readers to know and so briefly mention in the paper.

Detailed comments by page:

P10 top: “In line with our experimental instructions, IRIs was on average proportional to the number of samples that were generated before the nearest animal name to the sampler’s position changed.” – This further explanation of the relation between the sampling process and IRIs is very good to include, but it made me realize that I was unclear on another aspect of the sampling method, namely that samples are drawn from abstract points in the semantic space, *not* (solely) from points corresponding to particular words. This is clear for the time distribution in the first task, but perhaps should be filled in more for the semantic search task, e.g. that somehow the mind is sampling abstract semantic representations and only when these get close enough to the location of an actual word in the space is that word produced. It is also not clear to me in the above sentence what part of the instructions is meant here, and how the instructions would affect the relationship between (human-generated) IRIs and (model-generated) number of samples—clarify.

P10 top: “Further details about these assumptions and an exploration of an alternative in which IRIs are related to the distance between samples are given in Appendix A.” – this actually appears at the end of Appendix B, not A. It is good to see this comparison with an alternative cognitive model as a robustness check, however the comparison is somewhat concerning, because it shows that the results are *not* robust to this small change in the model, which makes HMC outperform MC3 and RWM (but not outperform MC3 and RWM in the original model in Table 1). This should be discussed more, to help readers decide what the implications of this result are for the overall generalizability of the results and conclusions of the paper.

P11 Discussion: “Across the two tasks, these results indicate that MC3 better described the human data compared to RWM and HMC.” – does this mean averaging/combining across the two tasks? This seems misleading, given that RWM (slightly) outperformed MC3 on animal naming.

P12 Table 2: It is not clear to me what the “MC3 no” line in the table represents? It doesn’t seem to correspond to “human data” as mentioned in the table caption, which I expected. Clarify.

**Have the authors made all data and (if applicable) computational code underlying the findings in their manuscript fully available?**

Reviewer #1: Yes

Reviewer #3: Yes

PLOS authors have the option to publish the peer review history of their article (what does this mean?). If published, this will include your full peer review and any attached files.

Reviewer #1: No

Reviewer #3: No

Figure Files:

Data Requirements:

Reproducibility:

References:

---

## [Editor Report · Decision Letter 2]

16 Jun 2022

Dear Dr. Zhu,

We are pleased to inform you that your manuscript 'Understanding the Structure of Cognitive Noise' has been provisionally accepted for publication in PLOS Computational Biology.

Best regards,

Samuel J. Gershman

Deputy Editor

PLOS Computational Biology

---

## [Editor Report · Acceptance letter]

11 Aug 2022

PCOMPBIOL-D-21-01984R2 

Understanding the Structure of Cognitive Noise

Dear Dr Zhu,

I am pleased to inform you that your manuscript has been formally accepted for publication in PLOS Computational Biology. Your manuscript is now with our production department and you will be notified of the publication date in due course.

With kind regards,

Olena Szabo
